# Reflexive Views of Virtual Communities of Practice among Informal and Formal Caregivers of People with a Dementia Disease

**DOI:** 10.3390/healthcare12131285

**Published:** 2024-06-27

**Authors:** Sandra Lukic, Connie Lethin, Jonas Christensen, Agneta Malmgren Fänge

**Affiliations:** 1Department of Health Sciences, Lund University, 221 00 Lund, Sweden; connie.lethin@med.lu.se (C.L.); agneta.malmgren_fange@med.lu.se (A.M.F.); 2Department of Social Work, Malmö University, 205 06 Malmö, Sweden; jonas.christensen@mau.se

**Keywords:** digital education, knowledge, KnowDem project, learning, professional development, thematic analysis, workshops

## Abstract

Knowledge seems to mitigate the consequences of dementia and new educational strategies are required. This study aimed to qualitatively explore the reflexive views and experiences of virtual Communities of Practice (vCoP) among informal and formal caregivers of people with dementia and explore vCoP as a tool for learning and knowledge development. Data were collected in a sequence of virtual workshops and analyzed and synthesized using thematic analysis. For the informal caregivers, one main theme emerged: *Learning and support*, comprising three subthemes: Strategies for learning; Creating emotional support; and in need of professional support. Among formal caregivers, one main theme emerged: *Professional development*, comprising two subthemes: Sharing and gaining knowledge and Knowledge as a professional tool. vCoP and collaborative learning using an educational platform seem to support learning and professional development among informal and formal caregivers, respectively. As a collaborative, virtual activities seem to provide practical and emotional support and promote professional development; vCoP seem to have the potential to promote the resilience and sustainability of care. Further research is necessary to gain an understanding of the effects of Communities of Practice (CoP) and vCoP and their successful implementation in care practices as well as the potential of using CoP in continuing professional development, CPD.

## 1. Introduction

In Sweden, about 130,000–150,000 people are diagnosed with a dementia disease, and this number is expected to increase [1,2]. Dementia is a progressive disease and those diagnosed often have complex needs [3]. Informal caregivers are a valuable resource in care [3], but the need for help from formal caregivers increases over time [4]. Informal caregivers include partners/spouses, other family members, friends, and neighbors, while formal caregivers include trained staff, in the Swedish context, employed by the county councils or municipalities. Approx. 84% of people with dementia are living at home with support from an informal caregiver [5], while a small number of people live in care homes.

Knowledge among caregivers is considered an important measure to mitigate the negative consequences of dementia [6]. It has also been demonstrated that education positively affects knowledge, self-efficacy, and attitudes towards dementia and contributes to improvement in communication and behavior management. Also, education targeting formal caregivers increases functional ability and reduces behavioral and psychological symptoms of dementia (BPSD) in the person with the disease [7].

Collaborative reflection is a process of critical thinking involving cognitive and affective interactions between two or more individuals who explore their experiences to reach new understandings [2]. Co-reflection strengthens individual learning and constitutes a driving force that leads to organizational learning and knowledge creation [8]. Communities of practice (CoP) can contribute to creating meeting places for knowledge development [9,10]. CoP are defined as a group of people with a common interest or area of concern [9], stimulating group members to learn from each other through sharing knowledge and experiences. CoP build a collective knowledge base in each member that, when applied, improves their individual performance and can significantly contribute to solving the problem they were brought together to address [11]. However, there is a general shortage of time for professional development among formal caregivers when it comes to time for reflection within a team, and the sharing of knowledge and experiences seems to be scarce [12]. Informal caregivers may face restrictions in time and space to participate in collaborative learning activities. Accordingly, new methods and new meeting places providing education and enhancing/supporting learning are crucial, not the least during the recent COVID-19 pandemic, where virtual meeting places substituted physical ones [13].

Virtual CoP (vCoP) have recently been integrated into healthcare [14] to, for example, effectively facilitate professional and interprofessional learning, transcending the limitations of time and space and increasing opportunities for knowledge sharing [15]. Adding virtual (v) into CoP presupposes the use of the internet, and the vCoP must meet the same requirements and characteristics as CoP to qualify as a CoP, as described by Lave and Wenger [16]. A scoping review indicated that vCoP support knowledge acquisition and resilience for people with dementia and their formal and informal caregivers [17]. vCoP may also have the potential to alleviate isolation, especially in rural areas where educational opportunities are scarce [18]. However, there is a significant lack of knowledge related to the views and experiences of vCoP in a care and services context. Thus, this study aimed to explore reflexive views and experiences of vCoP among informal and formal caregivers of people with dementia diseases and explore vCoP as a tool for learning and knowledge development.

## 2. Materials and Methods

### 2.1. Study Design

This study had a qualitative design following vCoP over time in the form of repeated virtual workshops. This study is a part of the KnowDem project, aiming to investigate different outcomes of CoP for informal and formal caregivers of people with dementia.

### 2.2. Participants

This study included informal caregivers (n = 5) and formal caregivers (n = 8). They were recruited through dementia nurses (informal caregivers) as well as the researchers’ existing networks (formal caregivers) in three municipalities in the southern part of Sweden. The participants were selected based on their interest in participating in a vCoP and digital workshop over one year. They were people caring for people with dementia who could provide written and verbal informed consent to participate in the Swedish language and had access to a computer and internet connection. Excluded participants were those who did not speak Swedish and who did not have access to a computer and internet connection. With one exception, the informal caregivers had participated in dementia care courses arranged by municipalities or interested organizations. All formal caregivers were auxiliary nurses or nurse aids. They had formal education in dementia care and considered their knowledge to be average to advanced. Their sociodemographic data are presented in Table 1.

### 2.3. Procedure

Separate vCoP were established for informal and formal caregivers, respectively. Starting in December 2022, four virtual workshops with informal (n = 3–5) and, starting in March 2023, three virtual workshops with formal (n = 4–8) caregivers took place every third month, with the last one in September 2023. For all workshops, a licensed version of Zoom was used, Lund University (LU)-Zoom. The participants freely chose the place for the workshops (home, workplace, or other). The aim of the workshops was to provide meeting places over an extended time period of one year for the CoP and explore in a structured way the participants’ views and experiences.

To provide a background and set the scene for the vCoP and trigger discussions in the workshops, a digital platform, EARLYDEM, available online at https://earlydem.com (accessed 21 May 2024), was used by the participants in their spare time. The platform comprises short lectures focusing on early signs of mild cognitive impairment and dementia, followed by multiple-choice questions. The platform is based on microlearning, a pedagogical approach applied to digital education consisting of small learning units [19]. Thus, it allows the user to pause and revisit lectures and questions at their own pace.

Prior to the workshops, written information was e-mailed to the participants. The information comprised the purpose of the study, confidentiality, data storage and treatment procedures, and information stating that all participation was voluntary. Written informed consent was signed by all participants before the first workshop.

A discussion guide with five open-ended questions was designed to guide the workshops (Table 2). Three researchers participated in the workshops, one male (J.C.) and two female. The female researchers were both registered nurses, with one being a PhD student (S.L.) and one being an associate professor (C.L.). The male researcher had a PhD in Education. All had extensive experience in facilitating discussions and were trained in collecting qualitative data through digital media. All workshops started with establishing rapport and a sense of the group climate. To increase the quality of the data, the same discussion facilitator participated in the workshops, following the guide and asking probing questions to deepen the discussions. Each workshop lasted between 18 and 29 min (informal caregivers) and 18 and 31 min (formal caregivers), and the discussions were audio- and video-recorded. These measures contributed to ensuring the trustworthiness of the findings. 

### 2.4. Data Analysis and Synthesis

The discussions were audio- and video-recorded and transcribed verbatim. In order to ensure the trustworthiness of our findings, the transcribed text was read through several times by both the first and second authors separately with the purpose of creating themes and subthemes. In parallel, to gain a general sense of the tone and content of the discussions, the recordings were returned to during the analysis process. In order to strengthen the credibility of the findings, and following Braun and Clarke’s thematic analysis [20], the first (S.L.) and second (C.L.) authors separately coded the text and developed preliminary subthemes and themes. The codes were closely compared with each other and corresponded with each other. In an iterative process, themes and subthemes emerged. The first and second authors then discussed subthemes and themes together to reach a consensus. Thereafter, the last author (A.M.F.) (registered occupational therapist, associate professor) reviewed the findings, followed by further analyses and revisions. Each participant was coded to ensure confidentiality.

Quotes corresponding to the respective themes and subthemes supported the findings.

First, codes and preliminary subthemes and themes were generated from each workshop separately. Subthemes and themes were then compared across workshops to explore how the participants developed their views and experiences over time. Some subthemes and themes were found to be more or less stable over time, while some were elaborated on and rephrased in the latter workshops. In an iterative process, subthemes and themes from all workshops were synthesized into themes and subthemes illustrating development over time for each group. 

That is, during the data analysis and synthesis process, measures were taken to guarantee that our findings were trustworthy.

The Consolidated Criteria for Reporting Qualitative Research Checklist (COREQ) was followed to report the research.

## 3. Results

For the informal caregivers, in the data synthesis including all workshops, one main theme emerged: Learning and support, comprising of three subthemes: Strategies for learning; Creating emotional support; and in need of professional support. Among the formal caregivers, one main theme emerged in the data synthesis: Professional development. This theme comprised two subthemes: Sharing and gaining knowledge and Knowledge as a professional tool (Table 3).

### 3.1. Informal Caregivers

#### 3.1.1. Learning and Support

This theme describes the individual strategies the participants used to increase their knowledge about dementia to enable learning and knowledge sharing and how different types of support played an important role.

##### Strategies for Learning and Knowledge Sharing

The informal caregivers’ most important strategy when seeking knowledge about dementia diseases was participation in different group meetings with other informal caregivers. They joined different organizations to meet others in the same situation. Experiences about practical things in everyday life were shared, but they also expressed that they had changed their perception of the disease in discussions with others.

“Before I came to the support group for informal caregivers, I didn’t know the extent of the dementia disease, so I have gained a completely different insight into the disease.”(Informal Caregiver [IC] 2)

Even if the informal caregivers searched for knowledge through different media, such as podcasts and websites, meeting groups enabled them to share experiences. A combination of real-life and virtual meeting places was considered the best method for knowledge exchange and learning. Virtual meetings were a good complement to physical meetings since some informal caregivers sometimes had difficulties attending physical meetings.

“I would probably think both…Because I also need to meet people … The digital education I can accommodate … whenever it suits me, since I can’t go to all the meetings…it is nice to meet people and talk…both those who are good at the subject, but also…someone who is in the same situation, who has an understanding, who does not judge.”(IC4)

Sharing experiences with each other, even when situations differed, allowed for mutual support and understanding. Also, friends and acquaintances were sources of knowledge. It did not matter what type of dementia one’s relative had, some experiences were still the same. Digital education required self-discipline and initiative, and the informal caregivers appreciated educational platforms that were accessible and easy to find and understand. The EARLYDEM platform was perceived as a tool that enabled learning as it offered short lectures, opportunities for repetition, and flexibility in time, place, and pace. The informal caregivers also pinpointed the need for regular updates of digital platforms with the latest research findings.

##### Creating Emotional Support

Informal caregivers’ social needs, such as obtaining emotional support and understanding from other informal caregivers, were best catered for in real-life (IRL) meetings. Even if virtual meetings with others in the same situations could reduce feelings of loneliness and isolation, real-life meetings provided a better opportunity to make new acquaintances and mitigate social isolation. Informal caregivers expressed the difficulties in coping with the complex situation of caring for a person with dementia. Meeting other people in similar situations helped them to express their feelings and improved their wellbeing. A positive, emotional climate in the group was of great importance, not only to gain knowledge. Warmth and compassion, talking to others who understood, being each other’s lifeline, and fostering feelings of belonging were crucial. 

“You don’t feel alone, when you hear that someone else…can share their experiences…it might get a little easier, when it’s heavy…you talk a lot about dementia diseases. But it fills a large part of one’s everyday life here, these relatives who have this disease. It’s actually really hard to be a relative…”(IC2)

Informal caregivers had a desire for recognition, emotional support, and to be seen by formal caregivers, something they felt they lacked.

##### In Need of Professional Support

Support from professionals was important to gain knowledge. The informal caregivers perceived that the professionals had an educational role and turned to them with questions. Informal caregivers expected group meetings arranged by the municipalities to have a clear agenda and a professional as a moderator. Dementia nurses and others were crucial as it was difficult to know what to ask for regarding the disease and its consequences. Fact-based knowledge and not just knowledge based on experience was asked for.

“…important that they [professionals] who hold…have a basic knowledge that is high. That it is fact-based. There is a little difference between experience-based knowledge and fact-based knowledge… Someone who is more familiar with the disease and the trajectory than me.”(IC3)

The informal caregivers also expected the professionals to be proactive and raise issues that they could foresee would happen, e.g., issues related to safety and economics.

### 3.2. Formal Caregivers

#### 3.2.1. Professional Development

This theme describes how professional development was the main motivator for their participation in different learning and knowledge-sharing activities.

##### Sharing and Gaining Knowledge

Formal caregivers used several strategies for sharing and gaining knowledge. The most important and common one was sharing knowledge in groups of other formal caregivers, e.g., in interprofessional meetings arranged by their employers or by themselves to accommodate their own needs. In collaboration, they discussed different topics and situations and shared knowledge and experiences, enabling them to gain new and deeper knowledge about dementia. Due to the complexity of dementia care, experienced professionals were requested to serve as supervisors to their colleagues. Most importantly, supervision was an important part of the formal caregivers’ learning process.

“I support my colleagues in their documentation, and with that, we learn a lot, how to treat…how to treat and what activities should look like…it is good, to discuss the how, to make it clearer.”(FC2)

Digital support was used to enhance learning. Notes were taken while listening to digital lectures and discussed with colleagues later. The EARLYDEM platform was easy to use and seen as a tool that enabled learning due to its micro-learning approach and the possibility to select time, place, and pace.

“An education is good if you can see it another time. An education you can stop and resume, I appreciate that…that you can rewind if you misunderstand, that you can go back and hear it one more time.”(FC7)

##### Knowledge as a Professional Tool

Formal caregivers perceived knowledge about dementia as a necessity enabling them to perform their work. In fact, knowledge was considered a professional tool. Repeating what they already knew was useful, but they desired updated, relevant, and evidence-based knowledge. Having deeper knowledge about dementia diseases and their consequences enabled them to explain and understand behavioral changes in the disease trajectory, educate informal caregivers, and help them understand what people with dementia are going through. Knowledge increased formal caregivers’ motivation to work and increase their learning.

“Sometimes you can find yourself in situations that you think are little difficult to handle. But with increased understanding…you feel that you can influence your situation and make the patient feel good…you also feel motivated, when you can make a difference with knowledge.”(FC6)

Acquired knowledge strengthened the formal caregivers’ reflexive capacity as a professional and was perceived to improve the well-being of the person they cared for. Support from management to participate in education was crucial; however, it was perceived as being of low priority, especially in times of financial restrictions. They expressed being left on their own to search for relevant and validated digital education and guidance that was requested from employers.

## 4. Discussion

This study aimed to explore the reflexive views and experiences of vCoP among informal and formal caregivers of people with dementia and explore vCoP as a tool for learning and knowledge development.

Learning and knowledge development was considered an important tool for professional development among formal caregivers, and the opportunity for learning was a major motivating factor. The informal caregivers largely shared the same driving force but from the perspective of caring for a specific person around the clock. While the formal caregivers requested support from their employers to develop their professional knowledge, the informal caregivers most of all needed emotional support from others in the same situation, but also from formal caregivers. For both groups, collaborative learning and sharing knowledge seemed beneficial, and digital education supported learning.

To the best of our knowledge, this is the first study exploring the use of vCoP among Swedish informal and formal caregivers of people with dementia. In our study, the informal caregivers’ learning process was an important motivator for participation in the vCoP. People with dementia have multiple needs [21], and, as pointed out by our participants, different kinds of knowledge, skills, and support are necessary. In this context, group meetings were a source of knowledge and subsequent learning, but their significance went beyond that. That is, in their communities, the informal caregivers also offered each other emotional support, which was crucial for coping with caregiver challenges. Similar findings have been demonstrated by Romero-Mas [22], who showed that contact with other informal caregivers provides social support and reduces feelings of isolation. As also found by Zwaanswijk et al., 2013 [23] and Lethin et al. [24], our participants expressed that the later stages of the disease are challenging, with less contact with friends and acquaintances, and, thus, there is a lack of necessary emotional support. In this context, professional support facilitates care provision and improves quality of life (QoL) for the informal caregivers and the people with dementia themselves [5].

When it comes to digital education, both informal and formal caregivers were positive towards the concept, although from different perspectives. The informal caregivers appreciated digital education as they experienced difficulties leaving home due to home care situations. As pointed out by Romero-Mas et al. [25], vCoP free them from time and space limitations, offering social support and contributing to less stress. However, in line with Byungura et al. [26], who reported that the majority of health care managers consider that digital education improves knowledge and practice, a combination of digital and IRL education was put forward as the best model.

Both informal and formal caregivers experienced the digital education platform used as easily accessible and flexible. As also pointed out by Shail et al. [19], self-accessed, self-paced, and self-regulated learning, with the possibility to go back and forth, where previous results and performance are accessible, seems to keep engagement levels high through rehearsal content. Informal caregivers expressed a need for a moderator who could also answer questions regarding dementia and contribute their knowledge. This is in line with Gairin-Sallan’s [27] findings that a moderator seems to be a motivator for knowledge exchange. As indicated by Romero-Mas et al. [22], the moderator’s role is to facilitate knowledge creation by giving support.

For the formal caregivers in our study, professional development was the most important issue and motivator for collaborative learning, thus contributing to professional development. This is in line with Karaferis et al. [28], who found that increased knowledge, job training, and opportunities to take initiative and exploit resources are factors that influence motivation and work engagement among healthcare workers. Among our participants, the main source of learning and development of their professional knowledge and skills was knowledge sharing and reflection in professional or interprofessional groups. Similarly, Knipfer et al. [8] reported that reflecting in groups at work by exchanging experiences leads to improved understanding and helps to develop best practices and knowledge creation.

The findings of our study put focus on the need for continuing professional development (CPD). CPD is a process of ongoing education and development [29] to be able to provide high-quality and safe care [30] by enhancing the knowledge and skills of professionals. CPD is emphasized by the European Union (EU). In this context, it is important to note that collaborative thinking helps develop problem-solving abilities [31].

This study has strengths and limitations. The interviews/workshops were quite short. Dicicco-Bloom and Crabtree [32] recommend a minimum of 30 min for each interview, and, in our study, a longer duration of each workshop likely would have enhanced a closer exploration of the topics discussed. Here, the virtual format may have had an impact on the length, i.e., virtual interviews and discussions seem to be shorter, potentially due to “Zoom fatique” [33], interpreted by Carter et al. [34] as cognitive fatique in digital meeting situations. Despite this, the workshops generated rich data revealing different views and experiences.

Initially, the participants were introduced to the EARLYDEM platform. We anticipated that it would take time to learn the platform and review all lectures. Most probably though, the three months allocated between the workshops were too long and may have affected the group dynamics and screen interactions negatively [34]. Despite this, the participants returned to the workshops, except for when they were ill or had appointments that were not possible to reschedule, thus indicating that the vCoP were perceived positively. 

In this study, we chose to separate the informal and formal caregivers into different vCoP. We anticipated that they would differ in views and experiences and that, therefore, the discussion would benefit from separating the groups. Mixing informal and formal caregivers in the vCoP would most probably have generated other discussions; however, we would not have been able to identify each group’s specific views and experiences. We strove to reach the same number of participants for both groups; however, this was not possible. Similar to what Brodaty et al. [35] pointed out, the home situation for informal caregivers can change rapidly due to the health status of the person with dementia and the high level of care burden and, thus, they could not always participate. For the formal caregivers, the unavailability of substitute staff may have hindered participation in some instances, in turn affecting the attendance rate.

Our data were collected over one-year (informal caregivers) and nine-month (formal caregivers) periods, respectively. The subthemes and themes emerging from the data analysis were found to be more or less stable over time. During the last workshop in each group, no new themes and subthemes were discussed. However, they were slightly rephrased and elaborated on, which we interpret as a result of the discussions and reflections taking place during and between the virtual meetings.

## 5. Conclusions

As a collaborative, the sharing of knowledge and experiences seems to provide both practical and emotional support as well as professional development. vCoP seem to have the potential to increase resilience capacity among caregivers and, accordingly, the sustainability of care. Further research is, however, necessary to gain an understanding of the effects of CoP and vCoP and their successful implementation in care practices as well as the potential of CoP and vCoP for sustainable professional development and continuing education.

## Figures and Tables

**Table 1 healthcare-12-01285-t001:** Participants’ sociodemographic data.

Participants	Measurement Scores
Informal caregivers (n = 5)	
Female gender, n	3
Age in years, (min–max)	47–72
Education, years	12–16
Formal caregivers (n = 8)	
Female gender, n	4
Age in years, (min–max)	36–55
Education, years	12–16

n = number.

**Table 2 healthcare-12-01285-t002:** Discussion guide for informal and formal caregivers.

Discussion Questions
1. In what ways do you as an informal/formal caregiver gain new knowledge about dementia diseases and different forms of support? 2. Are the meeting places (meeting points with other informal/formal caregivers) where you share the experiences of others regarding dementia and various forms of support important? 3. What is the significance of that type of meeting place? Can you give examples? 4. What is important to you when you request training/education and support? 5. What reflections do you make regarding digital education and support that can facilitate learning, provide knowledge development, and contribute to developing your work situation or making it easier in your everyday life?

**Table 3 healthcare-12-01285-t003:** Themes and subthemes from the data syntheses for formal and informal caregivers.

Themes	Subthemes
**Informal caregivers**	
Learning and support	Strategies for learning and knowledge sharing Creating emotional support In need of professional support
**Formal caregivers**	
Professional development	Sharing and gaining knowledge Knowledge as a professional tool

## Data Availability

The data used in this study contained sensitive information about the study participants, and they did not provide consent for public data sharing. The current approval by the Swedish Ethical Review Authority (dnr 2021-05145) does not include data sharing. A minimal data set could be shared by request from a qualified academic investigator for the sole purpose of replicating the present study, provided the data transfer is in agreement with EU legislation on the general data protection regulation and approved by the Swedish Ethical Review Authority.

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
