# Peer review of "Reflexive Views of Virtual Communities of Practice among Informal and Formal Caregivers of People with a Dementia Disease"

_healthcare, 2024, doi:10.3390/healthcare12131285_

Round 1

Reviewer 1 Report

Comments and Suggestions for Authors

I believe that this paper would benefit from revision. The topic is of interest. Caregiving in the dementia setting is exhausting and long term so insights are needed.

The methodology is not well articulated. The study design mentions a qualitative study via digital workshops. It does not provide a structure to replicate this study elsewhere. Participant numbers are very small, and there is no mention in the methods as to whether saturation was achieved through analysis.

The discussion and conclusion are repeated in this version of the paper, so that was very confusing.

The conclusion is insubstansial and does not compel the reader to consider adopting this approach.

I believe that the paper could be strengthened by revisiting the methodology, and strengthening the conclusion to highlight the benefit the study has had on the caregiver population. 

Author Response

Please see separate word document.

Reviewer 2 Report

Comments and Suggestions for Authors

Please state the place of the study

How were participants selected?

Please provide inclusions and exclusion criteria for participants

Was data saturation discussed?

Please follow the COREQ (COnsolidated criteria for REporting Qualitative research) Checklist, submit it , and the state in the section that COREQ has been followed

Sections of discussion and conclusions have been repeated

Author Response

Please see separate word document.

Reviewer 3 Report

Comments and Suggestions for Authors

Thank you for allowing me to review this interesting research on virtual communities of practice among (in)formal caregivers of people with dementia. 

  • General comment: I wonder if the research question is really what was studied; when looking at the results the focus is not on the value of vCOPs for learning, but much more on the experiences of formal and informal caregivers of people with dementia regarding their access to, use of and wishes for virtual learning opportunities. Do the researchers recognize this?
  Abstract: could the authors specify how data was collected, probably just by adding the wordt ‘qualitatively’ would already help. In addition, if allowed, a structured abstract could really enhance readability of the abstract.    Introduction: In general, the flow of the introduction is lacking and it goes a bit fast for its readers. For example, only one sentence about collaborative reflection (line 42-43) is mentioned, so as a reader one expects this whole part will be about  that, yet then the next sentence is about COP. The explanation of COPs is very nicely portrayed. What is collaborative reflection? Why is this considered important in the introduction? Then in lines 51-53 all of a sudden collaborative reflection for formal caregivers is mentioned again. Whereafter only one sentence is used to suggest informal caregivers face restrictions in time and space; could the authors elaborate why this is? I would suggest expanding the introduction to more clearly portray formal and informal caregivers needs for learning, why this is important and then rolling into the final paragraph regarding the gap.  I think the structure in this paragraph is really lacking. Please improve.  Line 56: why is COVID-19 being refered to? Due to the necessity of digital meetings? Again, it is going to quickly for the readers.    Line 37-38: what is meant by 'education involving caregivers’ and which ‘consequences’ are refered to? This sentence is unclear.    Line 57: vCoP is mentioned here for the first time, without specification of what the v stands for.    The gap and aim of the research have been presented very clearly!   Materials and Methods: 2.2: please specify in- and exclusion of the participants 2.3: I am confused when reading the procedures, as it seems that informal and formal caregivers attended different workshops and were henceforth not participating together in vCoPs? Is this correct? Are the workshops equal to the vCoPs? What is the difference between a workshop and a vCOP and the interviews? And if the aim is group learning, why are modules provided to the individual participants that they can study in their own time and pace? And why the choice to not have interdisciplinary groups (informal and formal combined)? What was the aim and content of the workshops? Were the interviews performed after the workshops? Were these individual or group interviews? 2.4: Data-analysis has been presented clearly   How was trustworthiness of the research guaranteed?   Results: Why are the findings preliminary?  It is quite difficult to follow that both intermittent results and final results are presented. Perhaps tables 3 and 4 would be better suited in an appendix, as background information. Now these are presented as main results whereas acutally Table 5 is what the findings are really about.   The findings are nicely described from Table 5 onwards.   Discussion: in general it is a nice reflexive discussion, yet I think more clarity on the research question and methods will strengthen this further.  Comments on the Quality of English Language
  • General comment: adjust dementia disease to dementia
  • General comment: english proofreading is recommended for grammar

Author Response

Please see separate word document.

Round 2

Reviewer 1 Report

Comments and Suggestions for Authors

This is a much stronger paper. thank you for revising and strengthening the methodology in particular.

The Discussion section is also strengthened by the revision.

Author Response

Thank you for your kind response.

Reviewer 2 Report

Comments and Suggestions for Authors

I would like to thank the authors for considering the comments and changing the manuscript accordingly. The manuscript has been improved. It can be accepted

Author Response

Thank you for your kind response.

Reviewer 3 Report

Comments and Suggestions for Authors

Thank you for allowing me to review this revision. The authors have incoporated the reviewer comments profoundly. Only, I am still missing information on How trustworthiness of the research was guaranteed. i would expect to read about this in the methods section. 

Comments on the Quality of English Language

No comments

Author Response

Thank you for providing additional constructive comments to our manuscript. We have now added requested information about the trustworthiness of our findings. More specifically, you find our new text in red on lines: 129-130; 133-147 and 155-156 respectively.